

# Wind profiles and low-level jet structures over the coastal waters of Japan

Kazutaka Goto[1,2], Takanori Uchida[3], Keisuke Nakao[1]

[1]Sustainable System Research Laboratory, Central Research Institute of Electric Power Industry, Chiba, 270-1194, Japan
[2]Interdisciplinary Graduate School of Engineering Sciences, Kyushu University, Fukuoka, 816-8580, Japan
[3]Research Institute for Applied Mechanics (RIAM), Kyushu University, Fukuoka, 816-8580, Japan

*Correspondence to*: Kazutaka Goto (ka-goto@criepi.denken.or.jp)

**Abstract.** Accurate characterization of coastal wind conditions is essential for offshore wind energy development; however, atmospheric structures in Japan's nearshore regions remain poorly understood. This study analyzed year-long vertical light detection and ranging (LiDAR) observations at closely located onshore and offshore sites along the Aomori coast to clarify the differences in wind profiles and their seasonal and directional dependence. Offshore wind speeds showed strong correlations ($r > 0.8$) with onshore data, indicating that, although direct substitution is inappropriate, onshore observations can effectively serve as reference data for offshore extrapolation when using the measure–correlate–predict (MCP) method. Low-level jets (LLJs) were frequently observed in spring and summer, particularly when wind directions aligned with the coastline, with occurrence rates ≥*20 %* higher than in other seasons. Case analyses revealed that diurnal transitions associated with land–sea breeze circulation modulate vertical mixing and surface friction, promoting the development of LLJs. These results advance our understanding of nearshore boundary-layer dynamics and provide a basis for improving assessments of offshore wind resources, turbine designs, and LLJ forecasting strategies.

## 1 Introduction

The importance of renewable energy continues to increase due to constraints on fossil fuel supplies and the effects of global warming (Zecca and Chiari, 2010; REN21 2024). In the future, offshore wind power is expected to become one of the primary sources of electricity generation (IEA and Wind, 2022). Because wind-power generation is proportional to the third power of wind speed, wind conditions play a vital role in multiple aspects of offshore wind projects. For example, annual wind speeds must be incorporated during the planning stage of offshore wind power to ensure that suitable locations are selected and to accurately assessing project viability (Bailey et al., 1997). We must also understand real-time wind speed during the operational stage of power stations because it directly affects the variability and uncertainty of wind farm electricity generation (Ward et al., 2023). However, in turbine load design, wind speed, turbulence intensity, wind shear, and veer must also be considered (Lundquist, 2022). The spatiotemporal distribution of inflow wind conditions is important to consider when evaluating turbine wakes (Lundquist, 2022; Porté-Agel et al., 2020), and wind speed influences scheduling maintenance vessel departures for operation and maintenance (O&M) activities (Si et al., 2025). Various studies have been conducted to better elucidate offshore





wind conditions, particularly in Europe (Dörenkämper et al., 2015; Schulz-Stellenfleth et al., 2022; Wagner et al., 2019). These studies have helped refine methods for evaluating offshore wind conditions (e.g., the appropriateness of using the power law to represent wind speed distributions) and improve wind forecasting techniques.

In general, a power law describes the vertical distribution of wind speed, with the exponent determined by land use. However,
in offshore environments, wind speed does not always increase monotonically with height as expected, and in many cases, the power-law model is not applicable (Goto et al., 2025). The occurrence of low-level jets (LLJs) is a prominent example of this deviation (Dörenkämper et al., 2015; Schulz-Stellenfleth et al., 2022; Wagner et al., 2019). In coastal regions, land-sea interactions shape wind conditions, leading to complex boundary-layer dynamics. These interactions contribute to the development of phenomena such as LLJs. For example, Dörenkämper et al. (2015) used observational data from FINO2
(Deutscher Wetterdienst, n.d.) and found that the formation of stable stratification over the ocean plays a significant role in LLJ development. Wagner et al. (2019) used FINO1 data and argued that diverse and context−dependent mechanisms are responsible for LLJ formation. Various other coastal wind features have also been investigated. For instance, a reduction in surface friction can accelerate wind speed as wind flows from land to sea (Taylor, 1969). This acceleration also amplified the Coriolis force, resulting in a slight veering of the wind direction (Emeis et al., 2007). These findings exemplify the key features
of coastal wind behavior relevant to offshore wind power development. However, the strong site-specific nature of wind conditions has necessitated continued research in this field. For example, local topography and climatic conditions can influence the occurrence of LLJs in coastal regions (de Jong et al., 2024; Qiu et al., 2023; Soares et al., 2022).

The terrain and climate in Japan differ markedly from those in Europe, where most previous research has been concentrated. Furthermore, offshore wind projects in Japan are often characterized by shorter fetches owing to the limited extent of shallow
coastal seas, resulting in unique operational conditions (Renewable Energy Institute, 2021). Understanding the wind behavior under such distinct conditions is vital for advancing offshore wind energy in Japan; however, existing studies are limited in scope (Goto et al., 2025; Konagaya et al., 2021; Shimada et al., 2018). Shimada et al. (2018) explored the relationship between fetch length and wind-speed acceleration using two vertical-profiling LiDARs, and Konagaya et al. (2021) conducted statistical analyses of observational data from coastal land–sea regions. The authors investigated how wind direction, seasonality, and
other factors affected variations in the wind speed exponent, mean wind speed, turbulence intensity, and related characteristics. These studies offer valuable insights into coastal wind conditions in Japan. However, being primarily statistical, they do not capture event-based phenomena such as LLJs or monotonic shear, which affect offshore wind projects in other regions (Debnath et al., 2021; Schulz-Stellenfleth et al., 2022). Goto et al. (2025) used UAV observations to report the rapid development of stable stratification in nearshore areas and vertical wind profiles that the power law could not adequately
capture. However, the underlying mechanisms remain unclear.

To address this research gap, this study investigated atmospheric structures in Japan's coastal regions using observational data from vertically profiled LiDARs installed on both onshore and offshore platforms. This study aimed to advance offshore wind energy in Japan by improving our understanding of coastal wind behavior. Statistical analysis was performed to identify the differences between onshore and offshore wind speeds, seasonal and directional variations in vertical wind profiles, and the



occurrence characteristics of LLJs. Furthermore, we conducted detailed case analyses of selected LLJ events to clarify their formation mechanisms in the coastal environment.

## 2 Observations and methodology

### 2.1 Observation overview and setup

The vertical atmospheric structure was analyzed using observational data collected along the coast of Rokkasho in Kamikita
District, Aomori Prefecture, Japan. The data were obtained from observation sites developed under a national research project by the New Energy and Industrial Technology Development Organization (NEDO). Notably, the dataset contributed to the development of the Offshore Wind Measurement Guidebook (NEDO, 2023) published by the NEDO, which is regarded as a highly reliable observational resource.

The surrounding environment and instrument locations are shown in Fig. 1. This figure also shows the onshore and offshore
observation points used in this study. The distance between the two sites was approximately *1.6* km. The surrounding area is flat and free of tall structures or complex terrain, thus minimizing the influence of local topography. The elevation of the onshore site was approximately *8* m a.s.l., resulting in a minimal height difference when compared with the offshore site.

Details for each observation point are listed in Tables 1 and 2. Each site was equipped with a measurement mast and a vertical profiling LiDAR. It should be noted that the measurement heights of the ZX300M LiDAR installed offshore varied depending
on the observation period. The number of valid data points available for each height varied seasonally.

For both onshore and offshore sites, the vertical profiling LiDAR data consisted of *10*-min average values recorded at *10*-min intervals. Low-availability data segments were excluded from the analysis to ensure quality and consistency. An ultrasonic anemometer mounted on the measurement mast recorded the wind data at a sampling frequency of *10* Hz.






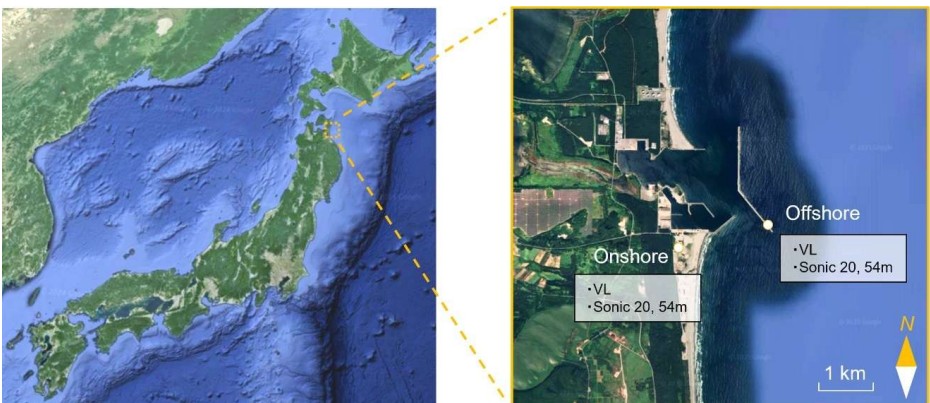

**Figure 1: Observation sites and the surrounding environment.**

VL indicates vertical light detection and ranging (LiDAR). Ovservation sites are the coast of Rokkasho in Kamikita District, Aomori Prefecture, Japan. Background map data: © 2024 Google (left) and © 2025 Google (right).

**Table 1: Onshore observational setup**

|  | Period | Interval | Data type | Height above land surface | Instruments |
|---|---|---|---|---|---|
| Vertical profiling LiDAR | 01/09/2023–31/08/2024 | ––– | *10*-min average | *40, 45, 50, 54, 58, 60, 70, 80, 90, 100, 110, 120, 130, 140, 150, 160, 180, 200, 250, 300* m | Windcube v2.1 (Vaisala; developed initially by Leosphere, France) |
| Ultrasonic anemometer | 01/09/2023–31/08/2024 | *10* Hz | *10* Hz | *20, 54* m | SAT-900 (Sonic Corporation, Japan) |







**Table 2: Offshore observational setup**

|  |  | Period | Interval | Data type | Height above sea surface | Instruments |
|---|---|---|---|---|---|---|
| Vertical LiDAR | profiling | 01/09/2023–30/11/2023 | *1* Hz | *10*-min average | *25, 53, 63, 100, 115, 130, 160, 250, 300* m | ZX300M (ZX lidars, UK) |
|  |  | 01/12/2023 –29/02/2024 |  |  | *25, 53, 63, 100, 115, 130, 160, 250, 300* m |  |
|  |  | 28/03/2024–31/05/2024 |  |  | *25, 53, 63, 100, 120, 130, 150, 160, 180, 240* m |  |
|  |  | 07/06/2024–31/08/2024 |  |  | *25, 53, 63, 86, 107, 120, 150, 180, 240* m |  |
| Ultrasonic anemometer |  | 01/09/2023–31/08/2024 | *10* Hz | *10* Hz | *25, 59* m | SAT-900 (Sonic Corporation, Japan) |

## 2.2 Low-level jet detection

LLJs are among the most prominent coastal atmospheric phenomena influencing offshore wind conditions. Their formation mechanisms are diverse and have been studied extensively in previous literature (Dörenkämper et al., 2015; Schulz-Stellenfleth et al., 2022; Wagner et al., 2019). However, no study has specifically addressed LLJs in the context of offshore wind energy development in Japan. Consequently, their occurrence, causes, and characteristics in Japanese coastal regions are poorly understood.

As highlighted by Hallgren et al. (2023), there is currently no consensus on a definition of LLJ, and the development of appropriate identification criteria is ongoing. The diversity in LLJ structures observed across regions and meteorological regimes has contributed to the lack of standardization. Thus, to investigate LLJs in this study, it was essential to first define them.

For initial LLJ identification, we referred to the detection criteria proposed by Wagner et al. (2019). In their approach, a LLJ is defined as a wind speed maximum located below a predefined height that exceeds the following local minimum aloft by at least *2* m/s$^{-1}$ and *25*%. If no local minimum above the maximum value was identified, the wind speed at the highest available measurement height was used as the reference minimum. These thresholds were designed to avoid false positives under low or high wind speed conditions. However, this study aims to understand offshore wind structures characterized by low-level wind speed maxima. To this end, we adopted the following modified detection criteria, which are schematically illustrated in Figure 2:

- A minimum peak wind speed of *4* m/s$^{-1}$, corresponding to the cut-in speed of typical wind turbines.

- A wind speed maximum that exceeds the following local minimum aloft by at least *10*% rather than *25*%.

These relaxed criteria allowed the inclusion of wind profiles that exhibited pronounced low-level maxima. Such profiles are of particular interest for understanding the wind resource characteristics in nearshore environments. LLJ detection was performed using *10*-min averaged vertical wind profiles derived from Doppler LiDAR measurements.

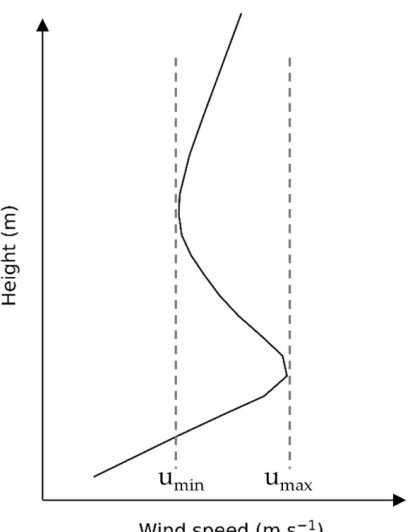

140    **Figure 2: Comparison of LLJ detection criteria: Wagner et al. vs. this study.**

Wagner et al. [11] use Eq. (1) for LLJ detection, whereas this study applies Eq. (2).

$$u_{max} - u_{min} \geq 2\ m\ s^{-1} \wedge u_{max} \geq 1.25\ u_{min}\ , \tag{1}$$

$$u_{max} \geq 4\ m\ s^{-1} \wedge u_{max} \geq 1.1\ u_{min}, \tag{2}$$

145

### 2.3 Atmospheric stability evaluation

Atmospheric stability is indispensable when analyzing wind conditions in the atmospheric boundary layer, and is often

quantified using the Monin–Obukhov length, L (Gryning et al., 2007). Several methods have been developed to estimate L,

including the eddy covariance method (Stull, 1988) and the bulk method (Grachev and Fairall, 1997). Although bulk methods

150    exploit vertical gradients in meteorological variables between two levels, these approaches may not accurately represent local

stability in oceanic regions, where vertical stratification is often non-uniform (Wagner et al., 2019). In contrast, the eddy

covariance method allows estimation of local atmospheric stability from flux measurements at a single point.





In this study, high-frequency data from ultrasonic anemometers installed onshore and offshore were used to calculate the Obukhov length using the eddy covariance method. Prior to analysis, a tilt correction was applied to the ultrasonic anemometer data using the planar-fit (PF) method (Wilczak et al., 2001). The regression coefficients ($b_0$, $b_1$, $b_2$) were determined daily using the linear relationship given by Eq. (2):

$$\bar{w}_m = b_0 + b_1 \bar{u}_m + b_2 \bar{v}_m \, , \tag{2}$$

where $\bar{u}_m$ and $\bar{v}_m$ are the mean horizontal wind speed components, and $\bar{w}_m$ is the mean vertical wind speed component, all of which are expressed in the instrument coordinate system. Following the approach of Wilczak et al. (2001), the regression coefficients were determined using $15$-min averaged data collected each day. After applying tilt correction, the deviations from the mean were calculated for each wind and temperature component. These deviations were then used to compute the friction velocity and $L$ as defined in Eqs. (3) and (4):

$$u_* = \left( \overline{u'w'}^2 + \overline{v'w'}^2 \right)^{1/4} , \tag{3}$$

$$L = \frac{u_*{}^3}{\kappa \frac{g}{T} \overline{w'T'}} \, , \tag{4}$$

where $u_*$ denotes the friction velocity. The constants $\kappa$ and g are the von Karman constant and gravitational acceleration, respectively, and T is the air temperature. Although the $L$ was calculated using the above equations, the offshore values of $\overline{w'T'}$ were extremely small and often exhibited spiky behavior. Therefore, we refrained from using the Obukhov length as a stability index and instead examined the vertical atmospheric structure using the friction velocity and the covariance of vertical velocity and temperature.

## 3 Offshore wind conditions near the coast of Japan

### 3.1 Comparison of onshore and offshore wind speeds

Based on the NEDO Wind Observation Guidelines (2023), wind speeds $120$ m above the onshore and offshore areas were compared for $16$ wind directions. The classification of wind direction was determined using $120$ m wind direction data from the offshore Doppler LiDAR. The correlation between onshore and offshore wind speeds varied significantly with wind direction (Fig. 3). High correlations were observed under sea–breeze conditions, whereas low correlations were observed under





land–breeze conditions. This trend is consistent with the findings of Konagaya et al. (2021) and can be attributed to differences in the surface roughness length. Specifically, under land breeze conditions, winds pass over inland areas with relatively high surface roughness, resulting in reduced onshore wind speeds. As the flow transitions offshore, where the roughness is lower, the wind speed increases, leading to a greater difference between the onshore and offshore wind speeds. In contrast, under sea-

breeze conditions, winds primarily travel over the ocean and coastal areas, both of which have relatively low surface roughness, allowing higher wind speeds to be maintained, even onshore. Consequently, the difference between the onshore and offshore wind speeds tends to be smaller. Furthermore, for wind directions other than land and sea breezes, particularly those parallel to the coastline (N and S), the correlation was lower than that for land breezes.

In summary, a comparison of wind speeds between onshore and offshore sites revealed that the data did not meet the criteria

specified in the NEDO guidelines (2023), indicating that onshore wind speed data should not be used directly as a substitute for offshore data. The meare-correlate-predict (MCP) method is commonly used to supplement missing wind observations by correlating data from nearby sites; for example, by using onshore data to estimate offshore conditions. In this context, a correlation coefficient of approximately 0.8 or higher is generally considered acceptable (Carta et al., 2013), and this criterion was satisfied in the present analysis. Therefore, for coastal locations with short offshore distances, such as those examined in

this study, the MCP method based on onshore wind data appears to be a valid and effective approach for estimating offshore wind conditions.

The vertical profiles of the normalized wind speed were averaged by wind direction and season (Fig. 4). In autumn and winter, and for land-breeze directions (WNW and WSW), the wind-speed differences between the onshore and offshore sites tended to be larger at lower altitudes. This can be attributed to the development of an internal boundary layer over the sea, driven by

a change in the surface roughness length from land to sea; that is, wind speeds in the internal boundary layer increase. In addition, during spring and summer, wind directions parallel to the coastline (N and S) and sea breeze (E) often exhibited LLJ structures in the vertical profiles, with wind speeds at lower altitudes exceeding those at higher altitudes. A detailed analysis of this phenomenon is presented in the following subsections.




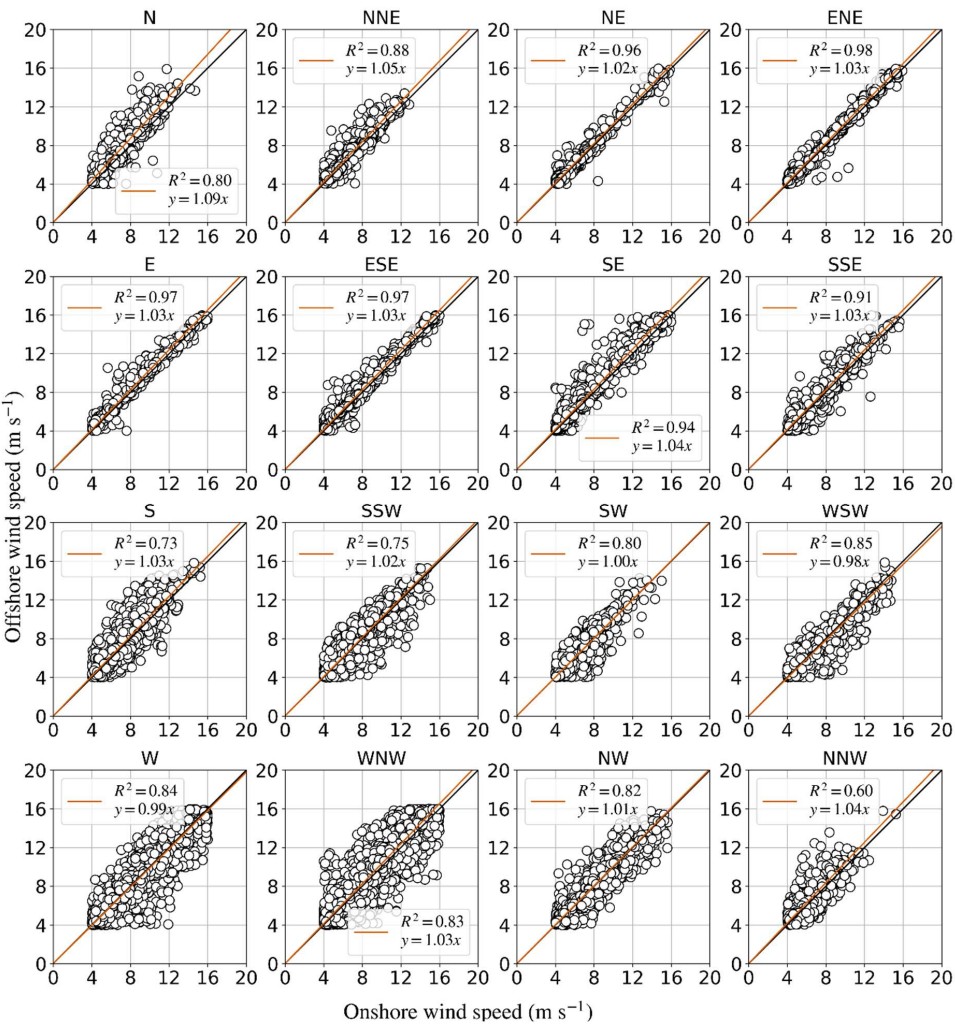

**Figure 3: Correlation between onshore and offshore wind speeds by wind direction at 120 m height.**

Wind direction labels follow meteorological convention (e.g., N = north, WNW = west-northwest).

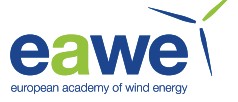
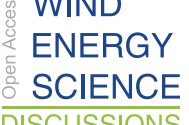
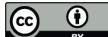

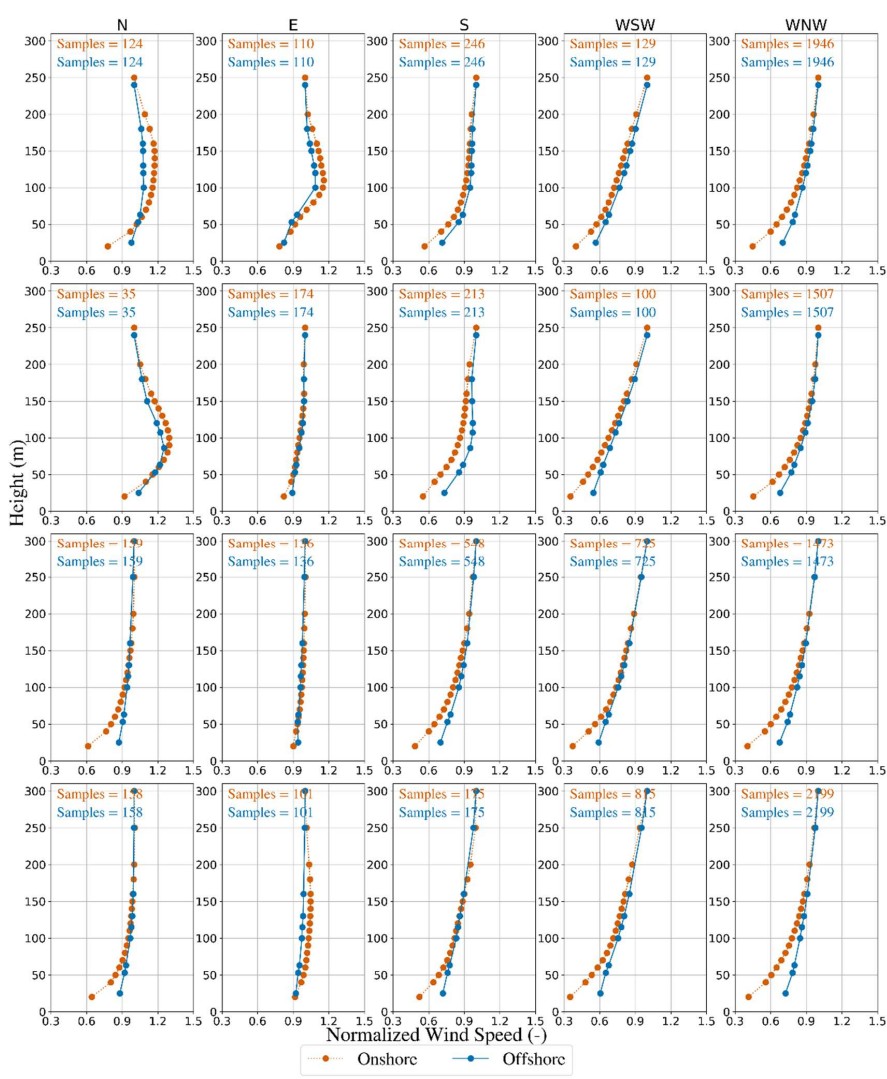


**Figure 4:** Vertical profiles of normalized wind speed averaged by wind direction and season.

Wind direction labels follow meteorological convention (e.g., N = north, WNW = west-northwest). Seasons are abbreviated as

MAM (March-May), JJA (June-August),SON (September-November), and DJF (December-February).






### 3.2 Frequencies and characteristics of LLJs

 Based on the detection method for LLJs described in Sect. 2.2, the occurrence frequencies of LLJs were calculated based on the wind direction and season (Table 3). The occurrence of LLJs exhibited a clear dependence on both wind direction and season, with particularly high frequencies observed in spring and summer. The high frequency of LLJs in spring and summer

is consistent with the results from European studies that focused on the North and Baltic Seas (Dörenkämper et al., 2015; Schulz-Stellenfleth et al., 2022; Svensson et al., 2016; Wagner et al., 2019). However, seasonal occurrence rates differ. For example, LLJs were reported to occur on *74–81 % of* days during spring and summer when using the FINO1 platform in the southern North Sea (Wagner et al., 2019). Svensson et al. (2016) found that LLJs occurred on *45%* of days in spring over the Baltic Sea. In contrast, this study indicates that LLJs are less frequent in Japan during these seasons. However, this conclusion

is based on a single observation site, and further measurements across multiple locations in Japan are required to validate this trend. In terms of wind direction, LLJs were found to occur more frequently in wind directions parallel to the coastline, specifically from the N and S.

**Table 3: Occurrence frequencies of LLJs by wind direction and season.**

|       | MAM            | JJA           | SON           | DJF           |
|-------|----------------|---------------|---------------|---------------|
| N     | 16.1 % (124)   | 42.9 % (35)   | 1.3 % (159)   | 0.6 % (158)   |
| E     | 18.1 % (110)   | 5.2 % (174)   | 0.0 % (136)   | 1.0 % (101)   |
| S     | 21.5 % (246)   | 28.7 % (213)  | 6.0 % (548)   | 0.6 % (175)   |
| WSW   | 7.0 % (129)    | 0.0 % (100)   | 4.3 % (725)   | 3.1 % (815)   |
| WNW   | 4.4 % (1946)   | 9.8 % (1507)  | 0.8 % (1473)  | 1.1 % (2199)  |

Values in parentheses indicate sample size. Wind direction labels follow meteorological convention (e.g., N = north, WNW = west-northwest). Seasons are abbreviated as MAM (March-May), JJA (June-August),SON (September-November), and DJF (December-February).

The hour-of-day distribution of LLJs was analyzed by wind direction during spring and summer, when they are most frequently

observed, to elucidate the mechanisms underlying their generation (Fig. 5). LLJ occurrence exhibits a strong dependence on the hour of the day. Furthermore, peak occurrence times varied with wind direction. For example, E tended to peak immediately





after noon, followed by S in the early evening. In contrast, N and WNW tended to peak at night, although N exhibited different trends in spring and summer. However, owing to the small sample size of N, this interpretation should be considered with caution. These findings suggest that the LLJ generation is closely related to the diurnal cycle. A more detailed discussion is

provided in Sect. 4.

The maximum wind speeds during the LLJ events are shown in Fig. 6. These wind speeds varied with the wind direction, with particularly high values observed in directions parallel to the coastline (N and S). However, the LLJ core wind speeds observed in this study were lower than those reported in Europe (Wagner et al., 2019) and the United States (Debnath et al., 2021). This indicates that LLJs over Japan may be characterized by relatively weaker wind speeds. However, it should be noted that this

result was based on a single site. Therefore, future studies using additional observational data across multiple coastal locations in Japan are necessary to evaluate the effects of LLJ characteristics on offshore wind energy development (Hallgren et al., 2023).

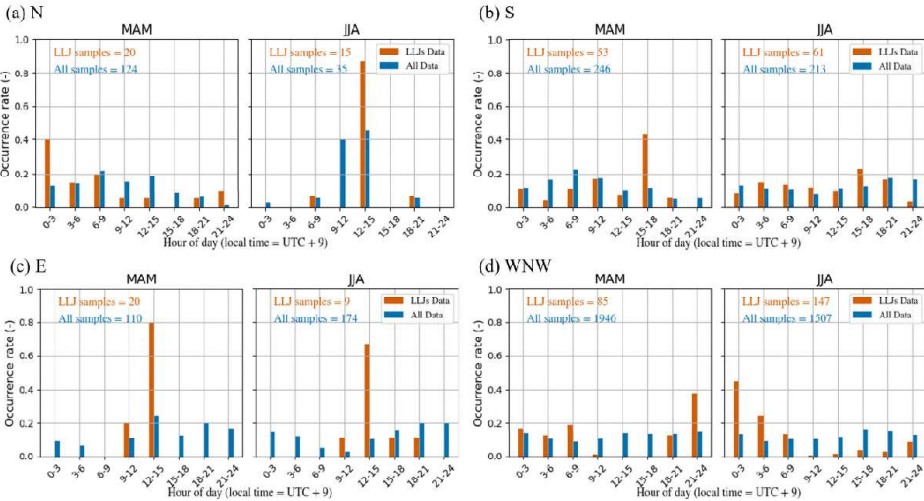

**Figure 5: Hour-of-day distribution of LLJ occurrences.**

Wind direction labels follow meteorological convention (e.g., N = north, WNW = west-northwest). Seasons are abbreviated

as MAM (March-May), JJA (June-August),SON (September-November), and DJF (December-February).

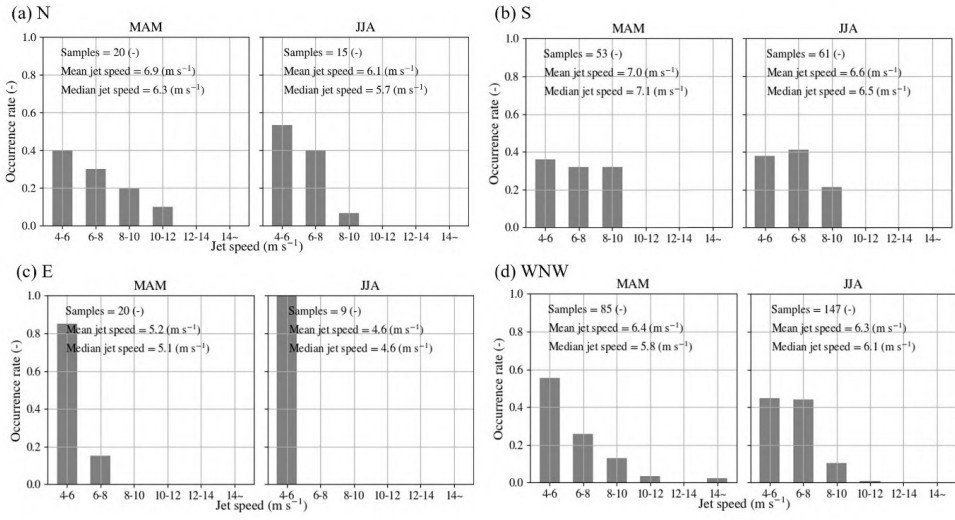


**Figure 6: LLJ core wind speeds by wind direction in spring and summer**

Wind direction labels follow meteorological convention (e.g., N = north, WNW = west-northwest). Seasons are abbreviated

as MAM (March-May), JJA (June-August),SON (September-November), and DJF (December-February).


**4 LLJ formation mechanism based on observational case studies**

In Sect. 3 it was revealed that LLJs exhibit not only seasonal dependence, but also strong associations with specific wind

directions and times of day. This directional and temporal regularity suggests a strong link between the LLJ formation and

diurnal atmospheric cycles. Several representative LLJ formation mechanisms have been proposed in the literature, most

notably those proposed by Blackadar (1957) and Holton (1967), both of whom attribute LLJ formation processes to diurnal



cycles. This section provides an overview of selected observational cases during spring and summer to explore the relationship between LLJ formation and diurnal coastal processes.

Time series data for wind speed and direction on May 2, 2024, are shown in Fig. 7, as a representative case of LLJs under easterly (E) and southerly (S) wind conditions. From night to early morning, the wind was a land breeze, gradually decreasing

in speed. This was followed by a shift to a sea breeze, representing typical sea–land breeze circulation. After this shift in the morning, an LLJ developed, as indicated by a wind speed inversion between the altitudes of *240*, *120*, and *53* m. In the afternoon, the wind speed fluctuated, and the wind direction gradually veered clockwise, transitioning from a sea breeze to southerly wind. Peak LLJ occurrence under E conditions occurred between 12:00 JST (local time = UTC + 9 h) and 15:00 JST, followed by a peak under S conditions between 15:00 and 18:00 JST (Fig. 5). Because Fig. 5 presents a statistical analysis,

this clockwise rotation of the wind direction is considered a typical feature. Before discussing these mechanisms in detail, we introduce another case of LLJ formation under land–breeze conditions. A time series of the wind speed and direction on July 11, 2024, is shown in Fig. 8. Nocturnal LLJ was observed under the land breeze. Similar to the previous case, wind speeds increased during the evening and night, along with a clockwise rotation of the wind direction. However, during this event, the wind speed decreased before fully transitioning to a northerly flow, and the wind direction began to return. These clockwise

rotations in wind direction are consistent with the mechanism described by Blackadar (1957). According to this theory, the transition from a daytime mixed layer to nocturnally stable stratification reduces the friction layer, allowing upper-level winds to become decoupled from surface friction. Consequently, the wind speed increases, and the wind direction rotates clockwise owing to the action of the Coriolis force.







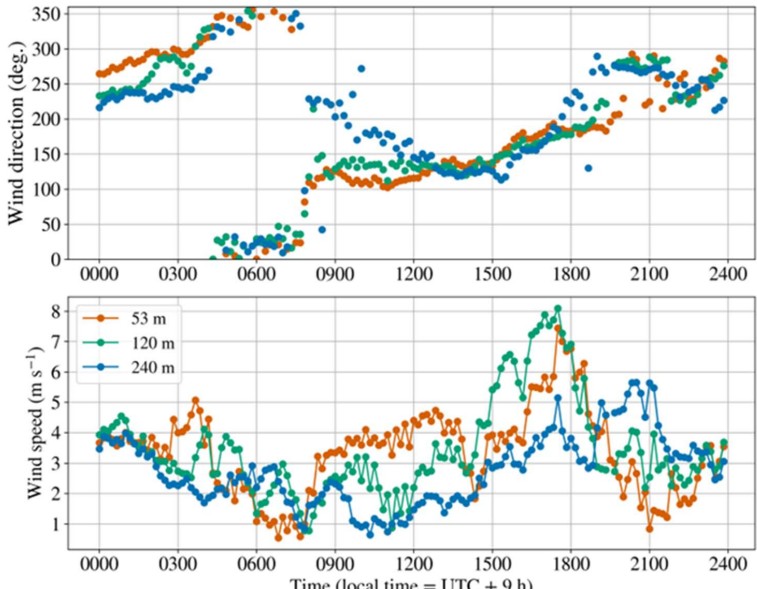

**Figure 7: Time series of wind speed and direction on 2 May 2024**

The legend indicates the observational height.




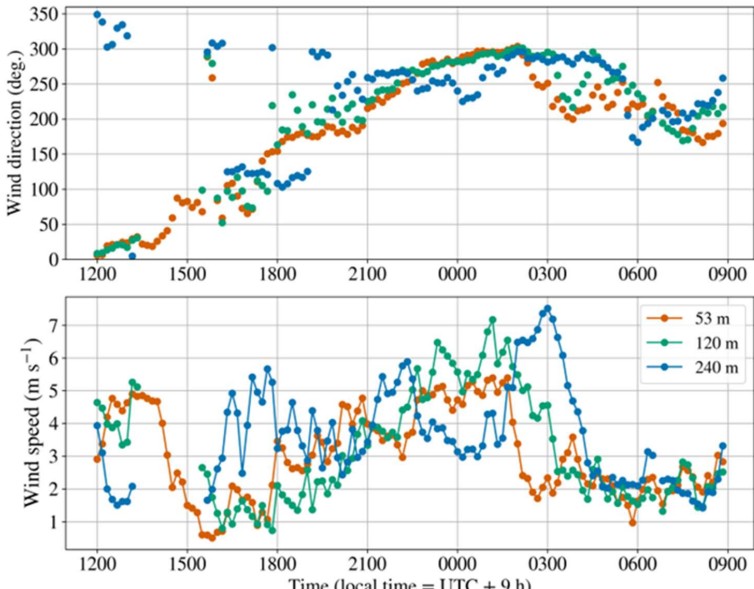

**Figure 8: Time series of wind speed and direction on 11 July 2024**

The legend indicates the observational height.


In this study, we interpreted the offshore LLJ formation mechanism observed on May 2 by integrating Blackadar's theory with the influence of sea–land breeze circulations. Figure 9 presents the time series of friction velocity $u_*$ and covariance of vertical velocity and temperature ($w'T'$). The calculation methods for these variables are described in Sect. 2.3. The friction velocity serves as an indicator of the depth of the friction layer, and the covariance of the vertical velocity and temperature represents

the strength of the thermally induced vertical mixing. Both variables were derived from observations 20 m onshore and *25* m offshore. The friction velocity was greater over the onshore site than over the offshore site from nighttime to daytime, indicating a thicker friction layer onshore (Fig. 9). Furthermore, the covariance of the vertical velocity and temperature onshore was low during the night but increased significantly to positive values during the day. In contrast, the diurnal variation offshore was small, with values remaining low throughout the day. These results suggest that the daytime friction layer over the offshore





area was thinner than that over the onshore area. Therefore, when the wind direction shifts from a land breeze (nighttime to early morning) to a sea breeze (daytime) due to sea–land breeze circulation, the upper-level winds become decoupled from surface friction as the friction layer thins. This process leads to an increase in wind speed and a clockwise rotation of the wind direction, consistent with Blackadar's mechanism (1957). Although Blackadar's theory attributes stratification changes to nocturnal radiative cooling over land, the LLJs observed in this study are driven by wind direction shifts associated with coastal

sea–land breeze circulation. Additionally, the thermal gradient between onshore and offshore areas, which drives sea breezes, may accelerate them and promote the development of LLJs. This topic will require further research.


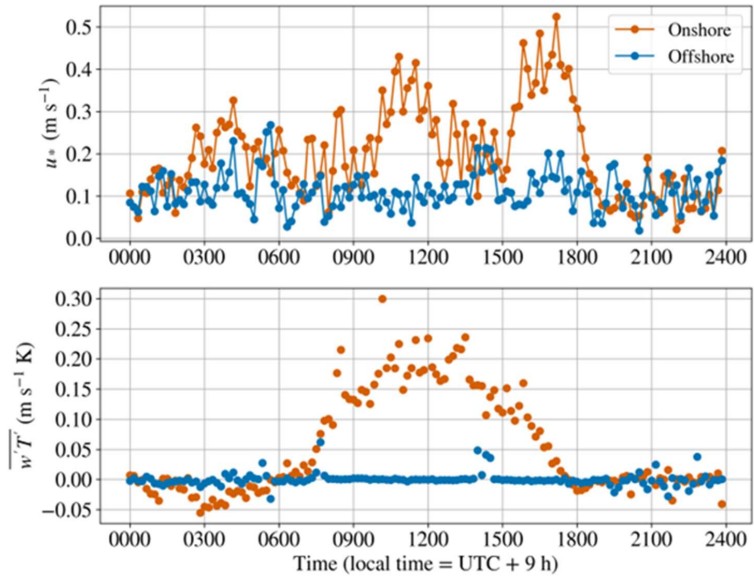


**Figure 9: Time series of friction velocity and covariance of vertical velocity and temperature at the onshore and offshore sites on 2 May 2024**

## 5 Conclusions

Wind behavior in Japan's coastal regions remains poorly understood. In this study, we aimed to elucidate this topic using LiDAR observational data obtained from both onshore and offshore installations. A comparison of wind speeds between onshore and offshore sites located *1.6* km apart indicated that nearby onshore data cannot directly substitute offshore wind conditions near the coast. However, application of the MCP method to onshore data could facilitate such a substitution. Year-round observations revealed that the occurrence of LLJs strongly depends on wind direction, season, and time of day. LLJs

were frequently observed when winds were parallel to the coastline, particularly during spring and summer, and occurred at specific times depending on the wind direction. Detailed analyses of individual LLJ cases suggested that wind direction changes associated with land–sea breeze circulation may contribute to LLJ formation. Specifically, transitions from land breeze





to sea breeze during the day and from sea breeze to land breeze at night suppressed vertical mixing and reduced the frictional influence on upper-layer winds, thereby facilitating LLJ development.

These findings help develop wind-condition assessments, designs, and operational strategies for wind turbines that account for LLJs, and to develop LLJ forecasting methods for offshore wind energy projects. The maximum wind speed observed in LLJs was close to hub height; the wind speeds themselves were relatively modest compared to those reported in other countries. Therefore, further quantitative evaluation is required to assess the impact of LLJs on wind energy development in Japan. In addition, characterizing LLJ behavior at other locations remains an essential task for future research.


**Code availability**

The code used for the data analysis was developed by the authors and is not publicly available.

**Data availability**

The observational data used in this study were obtained from the Kobe University/NEDO Mutsu-Ogawara Offshore Wind Observation Site ([https://mo-testsite.com/](https://mo-testsite.com/)) . The data are not freely available but can be purchased by anyone through the official website.

**Author contribution**

Kazutaka Goto conducted data curation, formal analysis, and visualization and contributed to the conceptualization, methodology, validation, and project administration. Takanori Uchida and Keisuke Nakao supervised the study and contributed to conceptualization, methodology, and project administration. Kazutaka Goto prepared the original draft, and all authors contributed to the review and editing of the manuscript.



**Competing interests**

The authors declare that they have no conflicts of interest.

**Acknowledgements**

We gratefully acknowledge the use of observational data from the Kobe University/MOC/NEDO Mutsu-Ogawara Offshore Wind Observation Site (https://mo-testsite.com/).

During the preparation of this work, the authors used ChatGPT in order to simplify expressions, correct typographical errors,
adjust formatting, and check the English language. After using this tool, the authors reviewed and edited the content as needed and take full responsibility for the content of the publication.

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
