# Peer review of "Wind profiles and low-level jet structures over the coastal waters of Japan"

_Wind Energy Science, 2025_

## Referee Comment (RC1)

Review of Goto, Uchida, and Nakao

Goto et al. provide a valuable analysis of offshore wind conditions at a site in Northeast Japan. In light of recent offshore wind interest in Japan and a relative lack of studies exploring wind conditions in the region, this study is timely and significant to the wind energy community. The authors' analysis of the offshore-onshore wind contrast as well as investigation of LLJ occurrence and mechanisms provide crucial information relevant to turbine design and farm planning at this site. I have a few comments related to the presentation of data and depth of analysis that will help to strengthen this paper, after which I believe it will be suitable for publication.

**Comments**:

1. Consider including more context for the current interest and future of offshore wind in Japan and relevance of the studied site in the introduction. In particular, a quick Google search revealed that most development interest appears to be on the West coast of Japan, while this study uses data from an East coast site. Further, is the 1.6km fetch (seemingly short) in this study representative of potential future offshore wind developments, or used purely for measurement convenience?

2. I feel that the presentation of data in figure 2 could be more effective by using a windrose or alternative display. In the current format of a single panel per wind direction showing onshore and offshore datapoints, the overall frequency and windspeeds at these directions is not readily apparent. However, accurate characterization of the winds at this site should include a clear comparison of the directions, seasonality, and magnitude. My specific recommendation is this:
    a. Include 4 windroses displaying mean 120m windspeeds, one for each season (MAM, JJA, SON, DJF), and each displaying both onshore and offshore winds.
    b. Report the R^2 correlation and the slope between onshore and offshore windspeeds as a function of wind direction on an additional set of windroses or as a line plot.

3. It is not clear to me why 5 wind directions are selected for display in figure 4, and later only 4 directions in Table 3 are selected. Please be explicit in why these particular directions are chosen for analysis. Why is WSW included first but excluded from the later analysis?

4. Related to comment #2, table 3 might be more illustrative if presented as a heatmap of LLJ counts. In the current tabular format, the table overemphasizes the importance of along-coast LLJs, when the absolute LLJ occurrence is in fact much

more prevalent in the dominant WNW wind direction as revealed by the data counts. A heatmap of LLJ count would allow relative comparison across a row (WD) or column (season) while remaining true to the absolute empirical data frequency.

5. L221-222: Given that LLJs occur frequently along-coast, have you ruled out a potential terrain influence such as has been observed along coastal California? Please comment on this mechanism and whether or not it may apply at this site.

6. L237: "with particularly high values [of windspeeds] observed in directions parallel to the coastline." I feel this statement is misleading, as 10-12 m/s windspeeds were also observed in WNW LLJs, and these WNW LLJs have a larger absolute occurrence than their along-coast counterparts.

7. Section 4 that analyzes LLJ case studies provides a well-written discussion of the influence of land-sea breeze circulation in causing frictional decoupling and subsequent inertial oscillation. However, I feel it could be strengthened and clarified with some additions:

   a. Arguments regarding rotation of the wind vector would be strengthened by including a hodograph of wind direction & speed at a representative altitude during each LLJ event for each of the two figures. You could further attempt to quantify the period of rotation to confirm that it is consistent with the Coriolis frequency and therefore the Blackadar mechanism.

   b. It is challenging to visually distinguish periods of LLJ occurrence from the wind speeds in figures 7 & 8. I recommend (a) shading the time period(s) of LLJ occurrence in both panels; and/or (b) including an additional panel that illustrates the full vertical windspeed profiles at a few times from LLJ onset to finish in each case study.

   c. The explanation of sea-breeze-induced frictional decoupling is consistent with the case studies shown, but the present analysis is not sufficient to characterize LLJs over the full statistical dataset. I recommend the authors expand this discussion to point out some other relevant characteristics of LLJ occurrence:

      i. The IO/sea breeze mechanism may also explain the relative prevalence of LLJs at along-coast directions as opposed to cross-coast winds. These less dominant along-coast wind directions may occur preferentially during inertial oscillation of the wind vector, hence their coincidence with LLJ activity.

      ii. Consider that the higher frequency of spring and summer LLJs likely corresponds with strong land-sea breezes in these warmer-land months, consistent with this mechanism.

iii. Land-sea breeze circulation is a daily occurrence, but LLJs are not.  It would be good to point out that this circulation is not a standalone *predictor* of LLJs, and to discuss other attributes of the offshore environment that may make LLJs more likely on certain days than others, such as large-scale temperature gradients, frontal systems (or a lack thereof), and other factors.

d. L267: The LLJ only appears between ~1400 – 1730, with windspeeds peaking around 1700, thus it would be more accurate to describe "peak LLJ" as occurring during the shift to southerly flow, not "E conditions".

e. It is worth pointing out that the second case study experiences an LLJ during WNW conditions, which appears to be the dominant wind direction at the site. Again, including a hodograph would aid in illustrating whether or not an inertial oscillation is present in this case.

**Minor/Technical:**

- L25: "assessing" → "assess"
- L39-40: clarify the region of FINO1 & FINO2 (North Sea, Baltic Sea)
- Figure 1 (right): the onshore dot is difficult to see on the similar-colored terrain
- Is the offshore measurement atop a breakwater, or floating?
- L136: "pronounced" → "less pronounced" ?
- Figure 4: The season labels are missing from the plot.